# Risk Assessments of Plant Leaf and Soil Mercury Pollution in Different Functional Areas of Changchun City

Jiafang Pan [1,†], Ming Chen [1,†], Zhe Zhang [1], Hongjie Zhang [1], Jing Zong [1], Zhaojun Wang [1,2,*] and Gang Zhang [3,4,*]

1. School of Environment, Northeast Normal University, Changchun 130117, China
2. State Environmental Protection Key Laboratory of Wetland Ecology and Vegetation Restoration, Changchun 130117, China
3. Key Laboratory of Vegetation Ecology, Ministry of Education, Northeast Normal University, Changchun 130117, China
4. Institute of Grassland Science, Northeast Normal University, Changchun 130117, China
* Correspondence: wangzj217@nenu.edu.cn (Z.W.); zhangg217@nenu.edu.cn (G.Z.); Tel.: +86-138-4480-1544 (G.Z.)
† These authors contributed equally to this work.

**Abstract:** Mercury is a global pollutant that has attracted widespread attention due to its persistence, concealment, and ease of migration. As the main setting for human production and life, urban ecosystems play significant roles in the global mercury cycle. Urban vegetation also plays an important role in regional mercury cycles. In this study, several common tree species in the green vegetation of Changchun were used as research objects to examine the mercury pollution characteristics of leaf litter and the surrounding soil. In this investigation, 100 sampling sites were set up in Changchun City to collect leaf litter and the surrounding topsoil. An RA-915+ Hg analyzer was used to determine the total mercury concentration levels. The results showed that content levels of mercury in plants ranged between 0.0133 and 0.1945 mg/kg, and different species displayed varying levels of mercury accumulation. For example, the content of mercury in *Pyrus ussuriensis* Maxim. was found to be 0.0755 mg/kg higher than that in the other examined subjects. It was also determined that the plant mercury concentration levels were the highest in the older industrial zones, while the lowest mercury levels were found in the new economic development zones. Furthermore, the plant mercury levels of the roadside areas were higher due to vehicle pollution discharge. Regarding the mercury pollution levels of the surface-layer soil, the soil of old industrial zones' surface layers had higher levels of mercury pollution than the other tested sites. However, no notable connection was observed between the mercury concentration levels of the vegetation and those of the soil. This study's results revealed that the mercury pollution of plants in Changchun City is not severe. The mercury levels in the industrial zones were graded as moderate, and those in the other regions were graded as low.

**Keywords:** mercury; green plants; soil; risk assessment

## 1. Introduction

Mercury is a persistent environmental pollutant; it has attracted global concerns due to its long-range atmospheric transportation around the world [1]. Mercury is regarded as a toxic element due to the potential threats it poses to the environment and to human health, its bio-accumulation in food chains, and its persistence in environments [2–4]. Mercury is often present in ecosystems in the form of $Hg^0$, $Hg^+$, and $Hg^{2+}$ [5,6]. Natural sources of Hg mainly include volcanoes, forest fires, and evaporation from water and soil surfaces. Meanwhile, anthropogenic emissions related to metal production, chemical industries, coal combustion, municipal and biomedical solid waste incineration [7–9], and the electronic, paper, pharmaceutical, and coal combustion industries are all major sources of Hg in the environment [10–12]. Over time and with the progression of industry development, the

application of mercury in various industries has increased, and large amounts of mercury-containing pollutants have entered the environments in which people live.

Cities play important roles in regional and global Hg cycling due to high population densities and intensive human activities. In urban areas where there are major sources of Hg, considerable effects are inevitably exerted on human health [13]. Urban ecosystems are sites for the consumption of both material flow and energy flow. The mercury contaminants in urban atmospheres mainly come from human production activities, such as fossil fuel combustion, non-ferrous metal smelting, cement production [12], steel production, transportation, and so on. Mercury from these various sources is discharged into the atmosphere and can migrate to surfaces (for example, soil layers), water bodies, vegetation, and so on through both dry and wet deposition.

Forest canopies can effectively trap mercury from the atmosphere since plants enable mercury accumulation by the adsorption of particulate bounded mercury (PBM) and reactive gaseous mercury on leaf surfaces, as well as the stomatal uptake of gaseous elemental mercury (GEM). It has been estimated that aboveground vegetation sequesters over 1000 tons of Hg from the atmosphere every year [1]. Vegetation is regarded as the missing sink in the global Hg mass balance [14]. The research regarding mercury absorption, storage, migration, and transformation in vegetation is an important component of the global mercury cycle. Urban vegetation also plays an important role in regional mercury cycles. In particular, the transport of mercury accompanied by "newborn withering" processes can be regarded as the process whereby urban atmospheric mercury pools are removed and purified. For example, Gustin reported that leaf litter is an important way for mercury to be deposited in forested areas, and leaf litter, together with the penetration flux, will vary according to different forest types. Forest tree leaves accumulate mercury during the growing season [14].

Atmospheric mercury is the main source of the mercury found in leaf tissue. However, the uptake of mercury by plant roots is limited [15]. Atmospheric Hg may be deposited into a forest canopy in gaseous and aerosol forms, and deposited $Hg^0$ and reactive gaseous Hg (RGM) may be taken up by leaf stomata [16]. Since vegetation covers nearly 80% of the land surface and the leaf area index is as high as 20 times the surface area, leaves can play an important role in the capture and circulation of many air pollutants [17]. Therefore, leaves are intermediate repositories of mercury in the atmosphere and also play an important role in aquatic environments through the global biogeochemical cycle and the movement of mercury between the atmosphere and lithosphere [18].

Soils are the largest repositories of mercury in the world [19]. Mercury mainly enters soil by being bound by organic matter, causing it to remain there for a long time. Mercury that has accumulated in soil can be slowly released into surface water and other media over time, a process that can last hundreds of years [12,20]. Moreover, mercury in the soil pollutes groundwater and agricultural products due to rainwater leaching effects; finally, it accumulates in the human body through the food chain, which, in turn, affects human health. Mercury is neurotoxic and teratogenic. Previous research has shown that even small amounts of mercury in the environment can cause serious harm to the human body [21], including corrosive bronchitis, interstitial pneumonia, and other diseases. Therefore, large accumulations of mercury in the soil will pose major environmental risks [22].

Litterfall is one of major components of energy flow and nutrient cycling in forest ecosystems. The global mercury flux through litterfall is estimated to range from 2400 to 6000 tons/year; it therefore constitutes the largest mercury flux in forest floors [23]. Recent studies have shown that soil mercury levels are indeed related to the litterfall mercury content. For example, approximately 90% of the mercury in surface soil layers is predominantly derived from the decomposition of litter in boreal forests [24]. However, few studies compare the soil mercury content and the mercury content of leaves. Far more research studies focus on speciated atmospheric mercury and atmospheric deposition in forest ecosystems [25].

Many research studies have been conducted on Hg concentrations in various cites [26,27], as well as on Hg exchange fluxes between environmental compartments in America and Europe [28,29], but little systematic research has been reported in China. Therefore, the objectives of this study are as follows:

(1) To examine the characteristics of mercury emissions in the leaves of broad-leaved forested areas in industrial cities (taking Changchun as an example), and to further explore the total content of mercury emissions in plants across one year through the mechanism of deciduous leaves, while exploring the differences in the mercury emission characteristics of different tree species.

(2) To investigate and analyze the factors affecting the enrichment of leaf mercury and the sources of environmental mercury pollution, providing a scientific basis for the prevention and control of urban mercury pollution.

(3) To assess the ecological risks of the mercury pollution levels in each region.

## 2. Materials and Methods

### 2.1. Study Area

Changchun (43°05′–45°15′ N, 124°18′–127°02′ E), the capital city of Jilin Province, is located in the northeastern plain of China and is known as "China's Detroit". Changchun is the province's financial, political, and cultural center [30,31] and is a rapidly growing city, with a population of 7.8 million people. The city covers an area of $20.5 \times 10^3$ km$^2$ [32]. The city's logistics-led tertiary industry group is mainly located in the northeastern section of the main city, and the industrial park region is mainly located in the southwestern section of the main city. The thermal industry is located in the southeastern part of the main city, where the release of coal particles into the atmosphere is an obvious source of mercury pollution. There is also a cement factory located in the northwestern section of the main urban area of Changchun [31].

This region is characterized by a subhumid climate, which features continental monsoons [33,34]. The weather patterns include windy and dry spring seasons (March to May); hot and humid summer seasons (June to August); dry autumn seasons (September to November); and long and cold winter seasons (December to February). The annual average temperature is 4.8 °C. The average annual precipitation is 567 mm, 70% of which occurs between the months of June and September [32].

The geology of the region consists of Quaternary alluvial–diluvial deposits. The natural soil types in the study area are mainly black, dark brown, and meadow soil, according to the classification and codes for Chinese soil (National standard, GB/T 17296-2009) [35]. The urban forest cover of the main urban area is 106.81 km$^2$, and it is mainly composed of coniferous and broad-leaved forested areas [36].

### 2.2. Sample Collection

#### 2.2.1. Sample Point Settings and Types of Sample Points

According to the Changchun City Master Plan (2010 to 2020), this study performed sample collections for each type of land use in the study area. The collection contained at least eight sampling points in October of 2019, with each point spot representing consistent land use. The following abbreviations were utilized in this study: W, warehouse land; R, residential land; B, commercial areas; M, industrial land; S, roads; A, public service land; and G, park land. The sample collection sites are listed in detail in Figure 1. The leaf forms mainly included *Populus tomentosa*, *Canadian poplar*, *willow*, *elm*, *Mongolian oak*, and other deciduous broad-leaved trees.

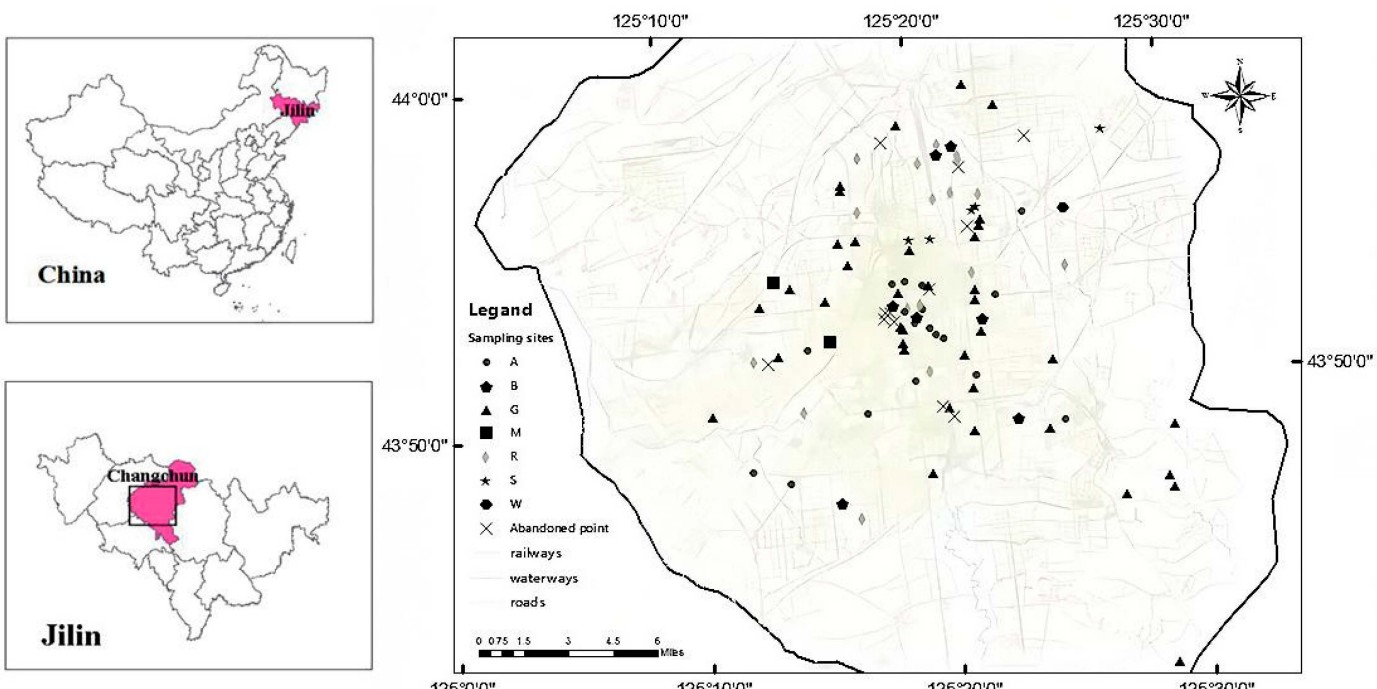

**Figure 1.** Study area and sampling points.

It was ensured that all the sampling points in this study were distributed as evenly as possible in the urban area of Changchun. A total of 100 sampling points were initially identified. However, during the field sampling process, 15 points were discarded according to the actual situation. Effective soil and deciduous samples were collected for the remaining 85 samples.

### 2.2.2. Collection of Deciduous Tree Samples and Soil Samples

In this study, a 100 m × 100 m sample square with the sample point as the center was set and GPS navigation was used for a five-point sampling process. Samples of ground-dried fallen leaves were collected around the sampling point and packed into self-sealing bags. At each sampling point, at least ten leaves of each tree type were collected. The deciduous samples were immediately sent back to this study's laboratory facility for air drying and were then effectively preserved for further analysis. After cleaning up the sundries on the soil surface, surface soil (1 to 5 cm) samples were collected in each sample square using a stainless-steel shovel. These were mixed together as the sample for one site, with each bag weighing approximately 500 g. Like the deciduous samples, the soil samples were immediately transported to the laboratory for air drying and were effectively preserved until further analysis.

### 2.2.3. Sample Pretreatments

The obtained deciduous samples were classified by tree species and crushed for preservation. Plants, dead leaves, stones, and sand were removed from the soil samples, which were then air-dried, ground, and sieved over 80-mesh nylon to remove rock fragments, human materials, and so on; they were then stored for analysis.

### 2.3. Sample Analysis

The Hg concentrations in the leaf litter and soil samples were measured using an RA-915$^+$ Hg analyzer with a PYRO-91 thermal decomposition accessory attachment (Lumex Inc., Moscow, Russia). The samples were thermally decomposed in an atomizer chamber at 750 °C, aided by catalytic action, and then the RA-915$^+$ analyzer was used to detect the Hg0. Each sample was analyzed three times and the results were averaged.

In this study, the instrument calibration curves covering the appropriate concentrations were confirmed by the soil standards (GBW07404, 590 ng/g) and peach foliage standards (GBW08501, 40 ng/g). The control standard samples were checked every ten samples. The instrument detection limit was 5 pg. for the solid samples. The precisions obtained from the 10 replicated determinations of standards were 3.47% for the soil and 2.83% for the peach foliage [1].

*2.4. Ecological Risk Assessment Processes*

2.4.1. Geo-Accumulation Index ($I_{geo}$)

The geo-accumulation index ($I_{geo}$) is used to evaluate the level of contamination of a particular metal in soil by assessing the concentration of metals above the baseline or background. In this study, the geo-accumulation index ($I_{geo}$) was calculated according to Equation (1), as follows [37]:

$$I_{geo} = \log_2 \left[ \frac{C_i}{K \times B_i} \right] \tag{1}$$

where $I_{geo}$ represents the ground accumulation index; $C_i$ is the measured concentration of mercury ($mg \cdot kg^{-1}$); K indicates the modified index (which is generally 1.5) in order to take into account the changes in the background values, which may have been caused by the differences in the rock characteristics in different locations; and $B_i$ denotes the municipal soil background value ($0.040 \ mg \cdot kg^{-1}$) [38].

2.4.2. Potential Ecological Risk Index (Er)

The potential ecological risk index (Er) was used in this study to evaluate the degree of potential ecological harm caused by the mercury in the soil and the atmosphere. The potential ecological risk index (Er) was calculated according to Equation (2), as follows [39]:

$$Er = Tr \cdot \frac{C_i}{C_0} \tag{2}$$

where Er is the potential ecological harm coefficient of the mercury; Tr represents the toxicity coefficient of the mercury, which was set at 40; $C_i$ is the measured value of the mercury content; and $C_0$ is the background value of the Hg [40]. The relationship between the potential ecological risk coefficient and the hazard degree of the Er was graded, as shown in Table 1.

**Table 1.** Criteria for the analysis of potential ecological risk.

| Er | <40 | 40 to 80 | 80 to 160 | 160 to 320 | >320 |
|---|---|---|---|---|---|
| Potential ecological risk index | Slight | Medium | Strong | Very strong | Extremely strong |

## 3. Results

*3.1. Concentration Levels of Mercury in Fallen Leaves*

3.1.1. Concentration Levels of Mercury among the Different Land-Use Types

The foliar Hg concentrations with different types of land use follow the sequence as shown in Table 2: M > S > A > G > R > B > W. The concentration of deciduous mercury in M (industrial land) was found to be the highest (0.0504 mg/kg) among all the land-use types, and the concentration of deciduous mercury in W was the lowest (0.028 mg/kg).

The table shows that the mercury content levels of the deciduous leaves varied significantly between the different land-use types in Changchun. Therefore, it was inferred that the mercury distribution patterns in Changchun were affected by human and industrial activities.

**Table 2.** Hg content levels in fallen leaves in the different land-use types in Changchun (mg/kg).

| Land-Use Type | Number of Effective Cases | Maximum | Minimum | Average | Standard Deviation |
|---|---|---|---|---|---|
| W | 2 | 0.0283 | 0.0277 | 0.028 | 0.0004 |
| B | 15 | 0.075 | 0.0241 | 0.0388 | 0.0117 |
| R | 51 | 0.0906 | 0.0189 | 0.0409 | 0.0136 |
| G | 114 | 0.0906 | 0.0155 | 0.041 | 0.0156 |
| A | 53 | 0.1945 | 0.0133 | 0.0463 | 0.0298 |
| S | 15 | 0.0868 | 0.0245 | 0.0487 | 0.0177 |
| M | 5 | 0.0822 | 0.0215 | 0.0504 | 0.025 |

**Note:** In the table, W is warehouse land; R represents residential land; B indicates commercial area; M is industrial land; S denotes roads; A is public service land; and G represents park land.

3.1.2. Mercury Concentrations in the Fallen Leaves of the Different Tree Species

As shown in Table 3, among the different tree species in the study area, the lowest concentration levels of mercury in the leaf litter were observed in *Populus alba* (0.0312 mg/kg), and the highest concentrations were found in *Pyrus ussuriensis* Maxim (0.0755 mg/kg). It was determined that the plant mercury levels reported in most areas, both domestically and internationally, are generally low. The range is concentrated within 10 to 50 ng/g, with an average of 24 ng/g. This study found that the mercury content levels of the main woody plants in Changchun were beyond that range. Therefore, based on those findings, it was confirmed that the mercury content levels in the plants in Changchun were high, indicating that the vegetation in the study area was polluted to a certain extent.

**Table 3.** Hg concentrations in the fallen leaves of the different tree species (mg/kg).

| Plant Type | Number of Effective Cases | Maximum | Minimum | Average | Standard Deviation |
|---|---|---|---|---|---|
| *Populus alba* | 6 | 0.0359 | 0.0261 | 0.0312 | 0.004 |
| *Quercus mongolica* Fisch. ex Ledeb. | 7 | 0.0413 | 0.0279 | 0.0342 | 0.0052 |
| *Populus tomentosa* | 28 | 0.1494 | 0.0201 | 0.0347 | 0.0233 |
| *Salix babylonica* | 1 | 0.0349 | 0.0349 | 0.0349 | 0 |
| *Populus pseudo-simonii* Kitag. | 32 | 0.062 | 0.0162 | 0.0353 | 0.0112 |
| *Salix matsudana* | 3 | 0.0433 | 0.0316 | 0.0357 | 0.0066 |
| *canadensis* Moench | 37 | 0.1945 | 0.0133 | 0.0363 | 0.0285 |
| *willow* | 41 | 0.0585 | 0.0197 | 0.0382 | 0.0092 |
| *Acer pictum* Thunb. ex Murray | 10 | 0.0594 | 0.02 | 0.0392 | 0.0128 |
| *Acer ginnala* Maxim. | 4 | 0.0663 | 0.0315 | 0.0429 | 0.016 |
| *Toxicodendronsuccedaneum* (L.) O. Kuntze | 9 | 0.056 | 0.033 | 0.044 | 0.0092 |
| *Prunus salicina* Lindl. | 4 | 0.0504 | 0.0427 | 0.0463 | 0.0037 |
| *Ulmus pumila* L. | 23 | 0.0868 | 0.0275 | 0.048 | 0.0145 |
| *Armeniaca vulgaris* Lam. | 7 | 0.0628 | 0.033 | 0.0499 | 0.0124 |
| *Catalpa ovata* G. Don | 3 | 0.0641 | 0.0436 | 0.0558 | 0.0108 |
| *Armeniaca mandshurica* (Maxim.) Skv. | 4 | 0.0637 | 0.0462 | 0.0567 | 0.0085 |
| *Malus baccata* | 12 | 0.0765 | 0.0267 | 0.0567 | 0.0129 |
| *Padus virginiana* 'Canada Red' | 6 | 0.0842 | 0.0436 | 0.0591 | 0.0165 |
| *Fraxinus mandshurica* Rupr. | 7 | 0.0906 | 0.0415 | 0.0645 | 0.0181 |
| *Padus racemosa* (Lam.) Gilib. | 6 | 0.0906 | 0.0478 | 0.0666 | 0.0156 |
| *Pyrus ussuriensis* Maxim. | 5 | 0.0896 | 0.0393 | 0.0755 | 0.0213 |

*3.2. Mercury Concentrations in the Soil*

3.2.1. Mercury Concentrations in the Soil

As shown in Table 4, the concentrations of mercury in urban soil in Changchun were between 0.0183 and 4.2967 mg/kg, and the skewness was 6.722, revealing a strongly positively skewed distribution. This skewed distribution indicated that some relatively highly Hg-contaminated urban soil existed in the collected samples. However, the skewness decreased to 1.708 when the log transformation was applied to the original Hg data. In this

investigation, the median (0.074 mg/kg) and geometric mean (0.1732 mg/kg) were used to explain the overall data because of the strongly skewed Hg distribution, and both were found to be above the background Hg level (Table 4). Therefore, an obvious accumulation of Hg was revealed to exist in the urban soil of Changchun. The soil of major urban areas in Changchun City was preliminarily influenced by exogenous mercury. It was found that the soil layers in different areas of the city were polluted by mercury to varying degrees.

**Table 4.** Description of the mercury content levels in the soil of Changchun City (mg/kg).

| Maximum | Minimum | Average | Median | Max/Min | Background |
|---------|---------|---------|--------|---------|------------|
| 4.2967 | 0.0183 | 0.1732 | 0.074 | 236.082 | 0.0400 |

As shown in Table 5, many other cities have carried out research on soil mercury pollution. Subsequently, through a process of comparison, it was concluded that the mercury levels in the surface soil of Changchun City were relatively low. The soil mercury content was not only lower than that of Beijing, which is seriously polluted by industry, but also lower than that of Chongqing, a coal-fired city. Furthermore, the soil mercury content in Changchun City was determined to be relatively low compared with Guilin and Guangzhou. However, the soil mercury content in Changchun was observed to be higher than that in such cities as Urumqi and Qingdao.

**Table 5.** Mercury content in the soil in other urban areas of China.

| City | Range | Average | Background | References |
|------|-------|---------|------------|------------|
| Chongqing | 0.060–3.881 | 0.319 | 0.040 | [41] |
| Beijing | 0.010–0.966 | 0.278 | 0.058 | [42] |
| Guangzhou | 0.013–12.231 | 0.614 | 0.157 | [43] |
| Taiyuan | 0.040–0.297 | 0.105 | 0.098 | [44] |
| Tibet | Unknown–0.056 | 0.026 | 0.021 | [45,46] |
| Urumqi | 0.012–0.176 | 0.062 | 0.055 | [47] |
| Nanjing | 0.041–8.090 | 0.043 | 0.025 | [48] |
| Ningbo | 0.010–0.565 | 0.103 | 0.143 | [49] |
| Guiyang | 0.010–7.030 | 0.222 | - | [50] |
| Guilin | 0.136–1.873 | 0.557 | 0.150 | [51] |
| Lanzhou | Unknown–0.117 | – | 0.150 | [52] |
| Zhaoyuan | 0.002–0.815 | 0.149 | 0.019 | [53] |
| Qingdao | 0.004–0.259 | 0.081 | - | [54] |

The mercury content of the surface soil in Changchun City was found to be within the global range of soil mercury content (30 to 100 ng·g$^{-1}$). However, vigilance is still needed to prevent the artificial introduction of mercury.

### 3.2.2. Values of the pH and Soil Organic Matter

The pH values of the soil samples ranged between 6.01 and 8.35, with an average value of 7.72, indicating that the alkaline condition of the urban soil in Changchun was relatively weak. The organic matter of the soil samples ranged from 11.24 to 140.30 g/kg, and the differences in pH and organic matter were not statistically significant ($p > 0.05$).

### 3.2.3. Soil Mercury Concentrations in the Different Functional Regions

This study found that the Hg concentrations according to the different types of land use followed the following sequence as shown in Table 6: M > R > A > G > S > B > W. The geometric mean of Hg in the industrial land (2.18 mg/kg) was obviously greater than that in the other six land-use types (0.06 to 0.28 mg/kg). This study implemented hypothesis testing to determine whether there were differences in the Hg values between the different types of land. The results showed that the differences in the Hg concentration levels between the industrial land and the other six types of land use were statistically significant

($p < 0.05$). For example, the differences between A and R and the differences between G and R were statistically significant ($p < 0.05$), while there were no significant differences between B, S, and W. These results imply that the type of land use had significant effects on the distribution of Hg in the urban soil of the study area.

**Table 6.** Soil Hg content levels in the different regions (mg/kg).

| Land-Use Type | Number of Effective Cases | Maximum | Minimum | Average | Standard Deviation |
|---|---|---|---|---|---|
| A | 18 | 0.5707 | 0.0217 | 0.0971 | 0.13463 |
| B | 7 | 0.1467 | 0.0190 | 0.0782 | 0.05154 |
| G | 38 | 0.2610 | 0.0183 | 0.0874 | 0.54540 |
| M | 2 | 4.2967 | 0.0713 | 2.1840 | 2.32117 |
| R | 17 | 2.0733 | 0.0310 | 0.2767 | 0.54144 |
| S | 5 | 0.1257 | 0.0537 | 0.0850 | 0.02750 |
| W | 1 | - | - | 0.0643 | 0.00723 |

The average values of the 7 functional areas displayed significant differences, with an F value of 25.798 and a corresponding probability value of 0.000.

As shown in Table 7 and Figure 2, there were significant differences observed between A and M, significant differences observed between A and R, and significant differences observed between B and M. Furthermore, significant differences in Hg content levels were found between G and M, G and R, and M and R. Significant differences were also observed between M and S and between M and W.

**Table 7.** Significance tests of mercury and soil organic matter.

| Land-Use Type (I) | Land-Use Type (J) | Mean Difference (I–J) | Std. Error | Sig. | 95% Confidence Interval | |
|---|---|---|---|---|---|---|
| | | | | | Lower Bound | Upper Bound |
| A | B | 0.0189 | 0.11285 | 0.867 | −0.2034 | 0.1210 |
| | G | 0.0097 | 06934.0 | 0.889 | −0.1268 | 0.1463 |
| | M | −2.0869 * | 017765 | 0.000 | −2.4368 | −1.7370 |
| | R | −0.1796 * | 0.08277 | 0.031 | −0.3426 | −0.0165 |
| | S | 0.0121 | 0.12090 | 0.920 | −0.2260 | 0.2502 |
| | W | 0.0328 | 0.24453 | 0.894 | −0.4489 | 0.5144 |
| B | A | −0.0189 | 0.11285 | 0.867 | −0.2411 | 0.2034 |
| | G | −0.0091 | 0.10439 | 0.930 | −0.2148 | 0.1965 |
| | M | −2.1058 * | 0.19403 | 0.000 | −2.4879 | −1.7236 |
| | R | −0.1984 | 0.11376 | 0.082 | −0.4225 | 0.0256 |
| | S | −0.0068 | 0.14390 | 0.962 | −0.2902 | 0.2766 |
| | W | 0.0139 | 0.25668 | 0.957 | −0.4917 | 0.5194 |
| G | A | −0.0097 | 0.05934 | 0.889 | −0.1463 | 0.1268 |
| | B | 0.0091 | 0.10439 | 0.930 | −0.1965 | 0.2148 |
| | M | −2.0966 * | 0.17240 | 0.000 | −2.4362 | −1.7571 |
| | R | −0.1893 * | 0.09082 | 0.008 | −0.3288 | −0.0498 |
| | S | 0.0024 | 0.11305 | 0.983 | −0.2203 | 0.2250 |
| | W | 0.0230 | 0.24075 | 0.924 | −0.4511 | 0.4972 |
| M | A | 2.0869 * | 0.17765 | 0.000 | 1.7270 | 2.4368 |
| | B | 2.1058 * | 0.19403 | 0.000 | 1.7236 | 2.4879 |
| | G | 2.0966 * | 0.17240 | 0.000 | 1.7571 | 2.4362 |
| | R | 1.9073 * | 0.17823 | 0.000 | 1.5563 | 2.2584 |
| | S | 2.0090 * | 0.19882 | 0.000 | 1.7074 | 2.4906 |
| | W | 2.1197 * | 0.19105 | 0.000 | 1.5464 | 2.6929 |

**Table 7.** *Cont.*

| Land-Use Type (I) | Land-Use Type (J) | Mean Difference (I–J) | Std. Error | Sig. | 95% Confidence Interval | |
|---|---|---|---|---|---|---|
| | | | | | Lower Bound | Upper Bound |
| R | A | 0.1796 * | 0.08277 | 0.031 | 0.0165 | 0.3426 |
| | B | 0.1984 | 0.11376 | 0.082 | −0.0256 | 0.4225 |
| | G | 0.1893 * | 0.07082 | 0.008 | 0.0498 | 0.3288 |
| | M | −1.9073 * | 0.17823 | 0.000 | −2.2584 | −1.5563 |
| | S | 0.1917 | 0.12175 | 0.117 | −0.0481 | 0.4315 |
| | W | 0.2123 | 0.24495 | 0.387 | −0.2701 | 0.6948 |
| S | A | −0.121 | 0.12090 | 0.920 | −0.2502 | 0.2260 |
| | B | 0.0068 | 0.14390 | 0.962 | −0.2766 | 0.2902 |
| | G | −0.0024 | 0.11305 | 0.983 | −0.2250 | 0.2203 |
| | M | −2.0990 * | 0.19882 | 0.000 | −2.4906 | −1.7074 |
| | R | −0.1917 | 0.12175 | 0.117 | −0.4315 | 0.0481 |
| | W | 0.0207 | 0.26032 | 0.937 | −0.4921 | 0.5334 |
| W | A | −0.0328 | 0.24453 | 0.894 | −0.5114 | 0.4489 |
| | B | −0.0139 | 0.25668 | 0.957 | −0.5194 | 0.4917 |
| | G | −0.0230 | 0.24075 | 0.924 | −0.4972 | 0.4511 |
| | M | −2.1197 * | 0.29105 | 0.000 | −2.6929 | −1.5464 |
| | R | −0.2123 | 0.24495 | 0.387 | −0.6948 | 0.2701 |
| | S | −0.0207 | 0.26032 | 0.937 | −0.5334 | 0.4921 |

**Note:** The data in the table are based on the measured average value, and the error term is the mean square (error) = 0.169. * The significance level of the differences between the average values is 0.05.

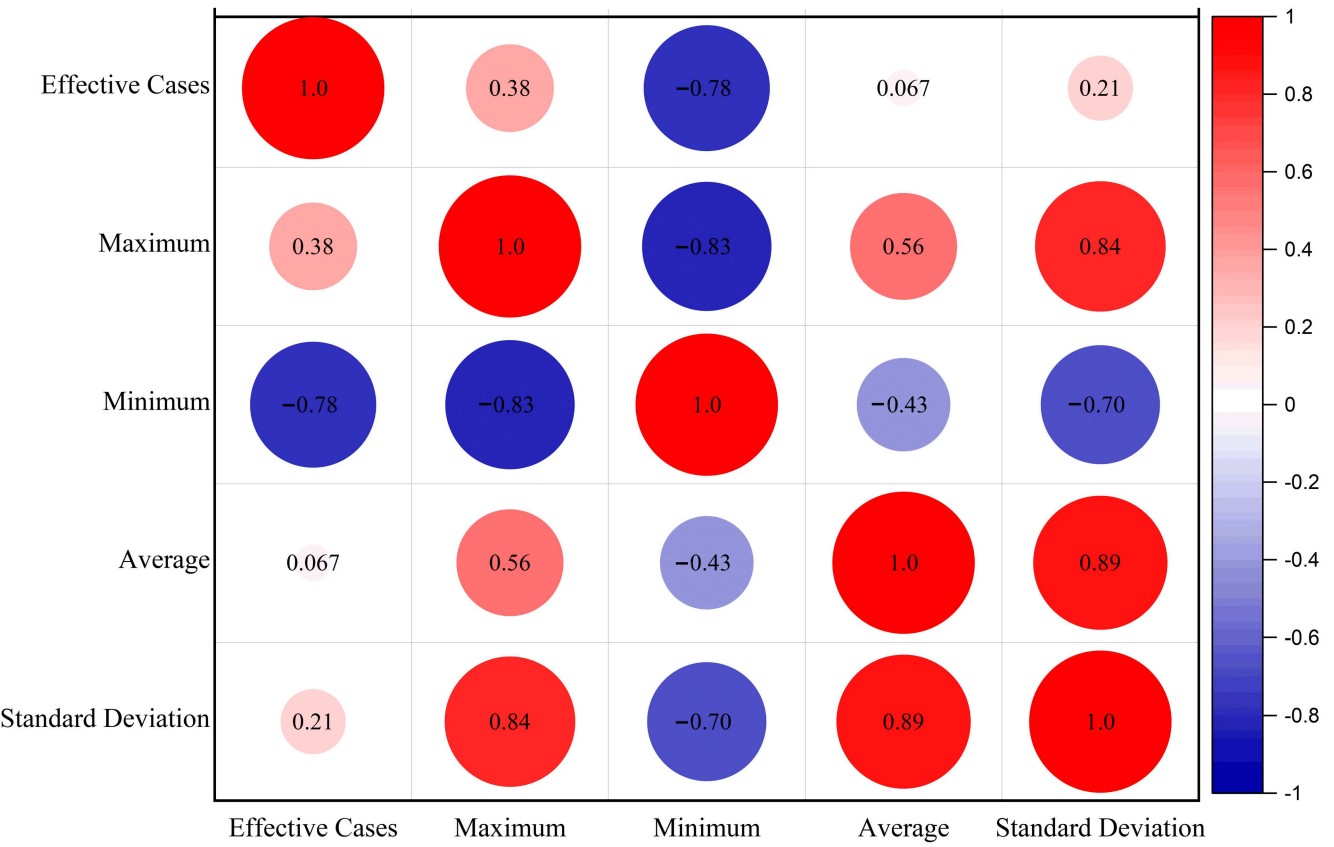

**Figure 2.** Pearson correlation analysis chart of the mercury content levels in the different land types.

Overall, the Pearson correlation coefficient was determined to be 0.035. That is to say, the input–output correlation coefficient was 0.035 and the *p* value of the bilateral testing

was 0.576, which was significantly greater than 0.05. Therefore, it was concluded that organic matter and Hg could be considered, and the correlation was not significant.

As shown in Table 8, the results of the Pearson correlation analysis indicated that, in the industrial land of the study area, the soil organic matter and soil total mercury content displayed a significantly negative correlation. For the land areas characterized by roads and transportation facilities, it was observed that the two had a significant positive correlation. Additionally, a significant positive correlation was found to exist between the public management and public service land areas and the green space and square land areas.

**Table 8.** Correlations between the mercury content levels and the soil organic matter.

| Land-Use Type | Number of Effective Cases | Pearson Correlation | Sig (2-Tailed) |
|:---:|:---:|:---:|:---:|
| A | 51 | 0.339 * | 0.015 |
| B | 18 | 0.124 | 0.624 |
| G | 114 | 0.235 * | 0.012 |
| M | 6 | −0.997 ** | 0.000 |
| R | 48 | −0.118 | 0.426 |
| S | 15 | 0.643 ** | 0.010 |
| W | 3 | c | |

**Note:** * means that the sig value is less than 0.05, ** means that the sig value is less than 0.01, in general, as long as it is reached *, it can be considered significant.

### 3.3. Ecological Risk Assessment of Soil Mercury Pollution

In this study, a geo-accumulation index ($I_{geo}$) was used to evaluate the mercury pollution levels of the soil samples, which not only considered the influencing effects of natural geological processes on the background value of the soil mercury, but also considered the influence of human activities on mercury pollution. The adopted method could accurately assess the mercury pollution levels of the urban soil in Changchun City. Generally, the pollution degree of heavy metals can be divided into seven levels, as follows: no pollution when $I_{geo} \leq 0$; mild pollution when $0 < I_{geo} \leq 1$; almost moderate pollution when $1 < I_{geo} \leq 2$; moderate pollution when $2 < I_{geo} \leq 3$; almost heavy pollution when $3 < I_{geo} \leq 4$; heavy pollution when $4 < I_{geo} \leq 5$; and severe pollution when the $I_{geo} > 5$. According to the results obtained in this research study, 60.0% of the sample points in Changchun City were polluted (index > 0). Among those points, only one point was found to be severely polluted. Moreover, 10.0% of the sampling points were moderately polluted, 50.0% of the sampling points were non-polluted to moderately polluted, and the remainder of the sampling points were not considered to be polluted.

The potential ecological risk index of the soil mercury in the study area was found to range from 18.3 to 4296.7, and 66% of the samples had a moderate or higher potential for ecological risks (Er > 40). Among those samples, a serious ecological risk was detected when the Er ranged up to 4296.7 > 320. The possible reason for this is that the sample site was close to the industrial concentration area of Changchun, and earlier activities and the current industrial activities have made the soil mercury concentration levels higher, resulting in a high potential ecological risk of regional soil mercury. In addition, 34% of the samples were found to have high potential for ecological risk, while 32% had medium potential for ecological risk; the remainder had low potential for ecological risk. In general, this study found that the mercury pollution levels of the soil in the Changchun District presented high ecological risk potential.

## 4. Discussion
### 4.1. Factors Influencing Mercury Concentration Levels in Leaves
4.1.1. Effects of the Different Land-Use Types on Leaf Mercury Concentrations

This study's analysis of the samples obtained from the same species growing in different locations showed spatial differences in the total mercury content of the plants,

with plants in some locations being heavily polluted. The mercury content levels were determined to be closely related to the environments in which they lived.

There are significant differences in the content of the same plant in different functional areas. In general, areas less affected by human activities have lower mercury levels compared to other regions. In areas with dense populations and many vehicles, mercury content can be affected by car exhausts. The wastewater and exhaust gas discharged from industrial production in industrial areas contain significant amounts of mercury, so the mercury pollution in industrial areas is more severe than that in other areas.

According to the data shown in Figure 3, the foliar Hg concentrations in M and S were higher than those observed in the other land-use types. This may be related to industrial coal burning and industrial leakage. It was found that the Hg in Changchun City mainly originated from coal burning and industrial activities. Lvyuan District is an important industrial production area in Changchun City. The wastewater and waste gas discharged during industrial production processes may contain large amounts of mercury, resulting in more serious mercury pollution in these functional areas. At the same time, the content of deciduous mercury in S was also relatively high. This may be due to the heavy traffic along the roads and the traffic flow, with automobile exhaust mercury becoming an important source of mercury in the plants of those regions.

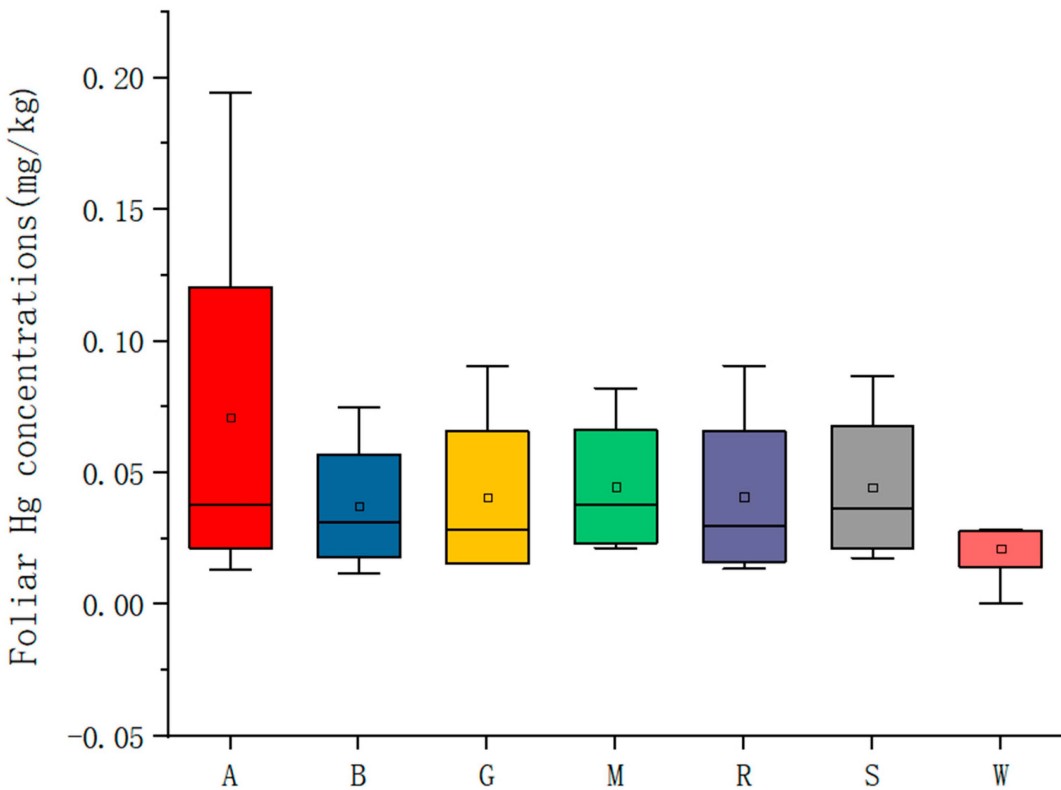

**Figure 3.** Foliar Hg concentrations of the different land-use types.

### 4.1.2. Effects of Different Plant Types on Deciduous Mercury Concentrations

Fallen leaves have long been used as passive monitoring tools in airborne mercury investigations and, more recently, as passive monitoring tools for atmospheric mercury. This is because trees are often found in landscapes, and it is cheaper and more convenient to measure such chemicals in leaves than to actively monitor them in the atmosphere.

The mercury in vegetation leaves mainly comes from the adsorption of atmospheric mercury oxide and particulate mercury on the leaf surface, as well as the absorption of zero-valent mercury in the atmosphere through the stomata. After zero-valent mercury is absorbed from the atmosphere, it can be converted into active $Hg^{2+}$ inside the leaves

and fixed by functional groups such as thiol (-SH) in the plant leaves to form chemically stable compounds (such as Hg (SR)2). Considering that the number of sulfhydryl groups (-SH) in vegetation leaves is much greater than the amount of free $Hg^{2+}$ in vegetation, they can provide sufficient binding sites for mercury fixation. Therefore, mercury shows a continuous accumulation trend during the growth cycle of plant leaves.

As shown in Figure 4, *Fraxinus mandshurica* Rupr., *Padus acemose* (Lam.) Gilib., and *Pyrus ussuriensis* Maxim were found to have high content levels of deciduous mercury. Therefore, those plant species can be utilized as green plants to absorb mercury from the air and improve the air quality.

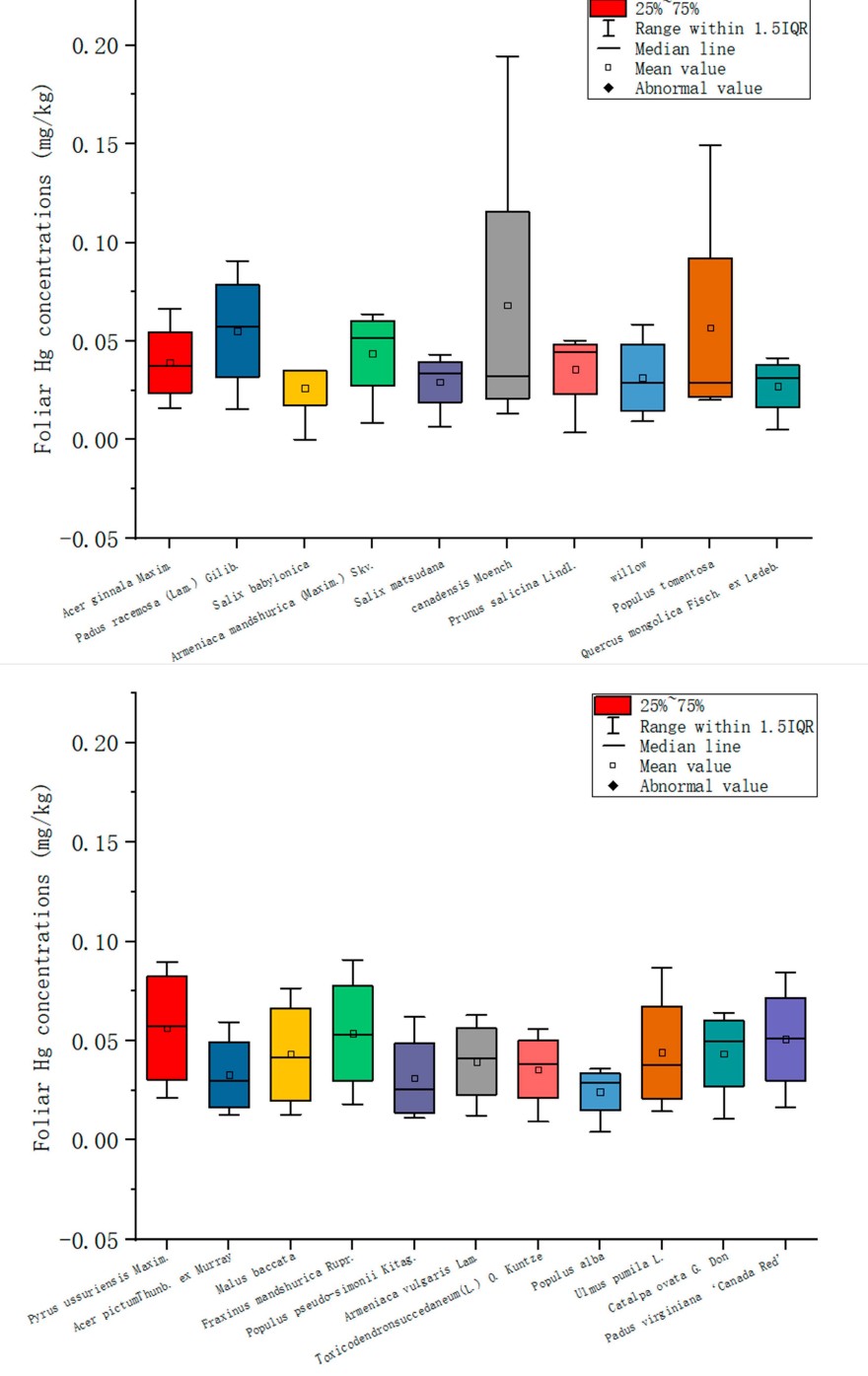

**Figure 4.** Foliar Hg concentrations in the different vegetation types.

### 4.2. Factors Influencing the Soil Mercury Content Levels

### Effects of the Different Land-Use Types on Soil Mercury Concentrations

Table 9 shows that the mercury content levels in the topsoil of the different functional land areas were significantly different, with the highest levels of mercury observed in the industrial land soils and relatively high levels observed in the roadside soils.

**Table 9.** Mercury pollution levels in the different land-use types.

| Land-Use Type | No Pollution | Light Pollution | Near Moderate Pollution | Moderate Pollution | Near Heavy Pollution | Heavy Pollution | Serious Pollution |
|---|---|---|---|---|---|---|---|
| A | 4 | 4 | 7 | 0 | 1 | 1 | 0 |
| G | 3 | 6 | 20 | 6 | 2 | 0 | 0 |
| R | 0 | 6 | 6 | 3 | 0 | 0 | 2 |
| W | 0 | 0 | 1 | 0 | 0 | 0 | 0 |
| S | 0 | 1 | 3 | 1 | 0 | 0 | 0 |
| B | 1 | 2 | 1 | 2 | 0 | 0 | 0 |
| M | 0 | 0 | 1 | 0 | 0 | 0 | 1 |

As shown in Figures 5 and 6, the differences in the soil mercury content levels in the different functional areas were similar to those observed in the plants. Therefore, it was suggested that the "three waste" emissions in the industrial areas were the main causes of the high mercury levels; the dry and wet sedimentation of the mercury emitted by coal-burning processes may be the main reason for the increased mercury levels in the topsoil [55]. On the other hand, the mercury in petroleum products tends to settle in nearby areas, along with vehicle exhaust emissions, resulting in higher mercury levels in densely trafficked areas.

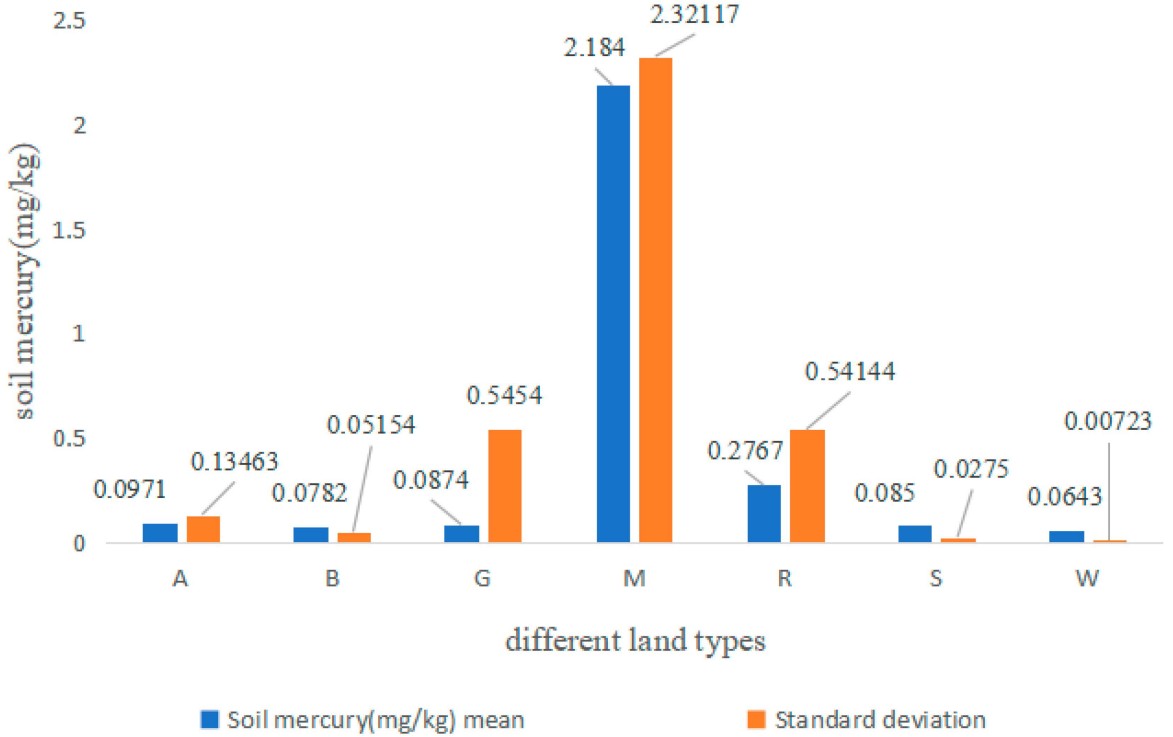

**Figure 5.** Soil Hg concentrations in the different land types.

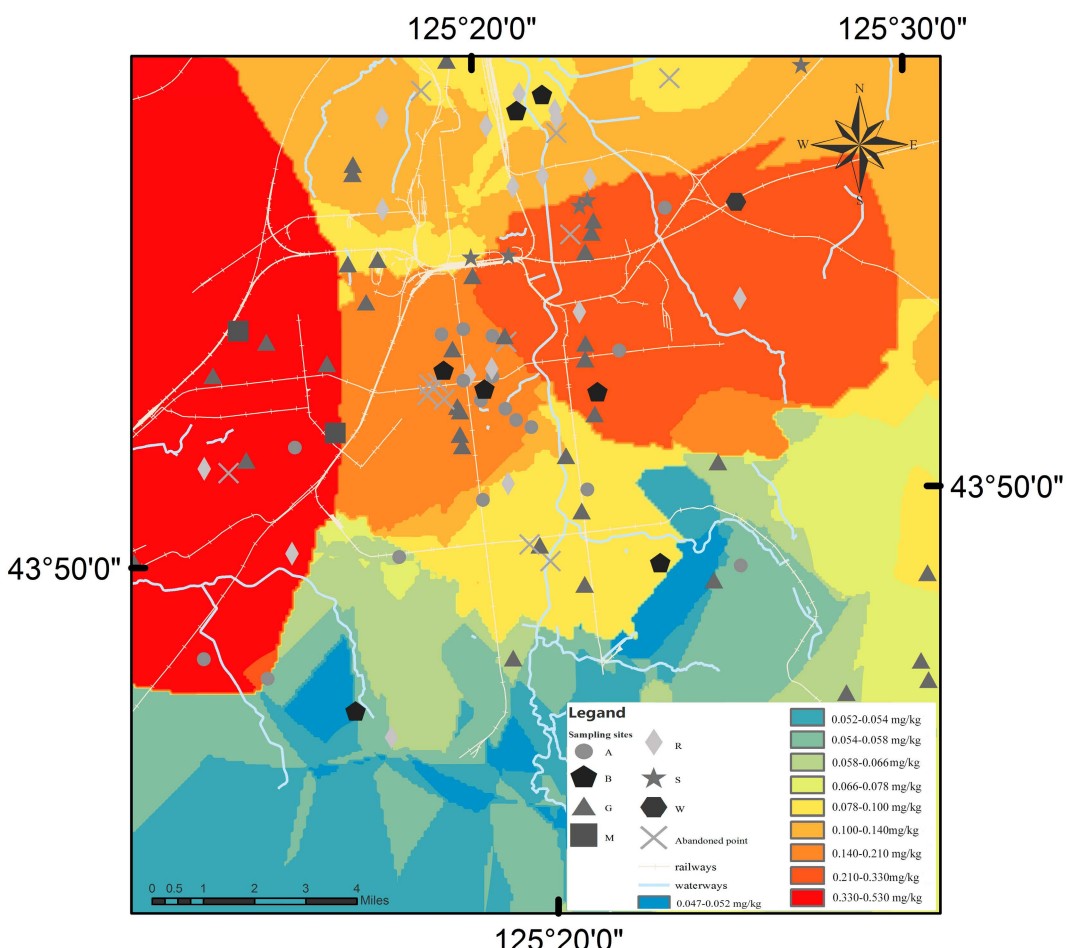

**Figure 6.** Spatial layout of the soil mercury concentrations in Changchun City.

### 4.3. Relationships between the Deciduous Mercury, Soil Mercury, and Atmospheric Mercury

The total mercury content levels of plants are affected by a variety of factors and are closely related to the environments in which the plants live. Among these factors, atmosphere–leaf interactions dominate the input and output of mercury by plants in the environment [55]. When the concentration levels of mercury in the air reach 20 ng·m$^{-3}$, plants absorb mercury from the air. Then, plants release mercury when the concentration levels of mercury in the air reach 2 ng·m$^{-3}$ [17], and those two concentration levels are close to those in the contaminated area and the background area, respectively. Changchun City has a certain degree of atmospheric mercury pollution, and its urban atmospheric gaseous mercury concentrations range from 4.7 to 79.6 ng·m$^{-3}$ [18]; the values have exceeded 20 ng·m$^{-3}$ in most areas over time, which is conducive to the absorption of atmospheric mercury by plant leaves.

In this study, the correlation analysis results using SPSS software showed that there was no significant correlation between the mercury content in the plants and the mercury content in the soil (not shown in the chart), indicating that the plants took up limited amounts of mercury from the soil. Plants can absorb mercury from the environment through roots and leaves, while mercury uptake from the soil is very limited [56]. This is mainly due to the fact that the uptake of mercury by plants from the soil is limited by conditions characterized by high levels of humus and organic matter in the soil, which can form inert compounds with the mercury, thereby affecting the mobility of mercury. This view has also been confirmed by numerous studies [20,21].

This study's correlation analysis results showed that the plants had a small correlation with the total mercury content in the soil, indicating that, with the exception of the roots,

the plants absorbed limited amounts of mercury from the soil [19,56]. Those findings were consistent with the results of related international studies. For example, the results of the studies conducted by Barghigiani [7] showed that pine leaves were positively correlated with atmospheric mercury levels, but pine leaves were not associated with root mercury levels. In addition, in the studies conducted by Lindberg [10], it was shown that less than 10% of the mercury in the fallen leaves each year could be absorbed by the roots. Based on the sap of *Swedish cloud shirt* and *pine xylem*, Bishop [20] determined that approximately 10% of the total mercury and approximately 3% of the methylmercury in deciduous leaves can be considered to be absorbed by the roots. Therefore, the previous research findings suggest that soil is not a source of leaf mercury, and that the atmosphere acts more on plants than on the soil. In addition, the root systems of arbor plants are large and extend to different levels of the soil, meaning that there is a wide absorption range of mercury in the soil environment. Consequently, this study's collected soil samples may have been quite different from the soil environments in which the plants were located [21].

Changchun has a long heating period and uses large amounts of coal, which have certain impacts on the atmospheric mercury concentration levels and accumulated soil mercury content levels. Since the main woody plants stop growing during the heating period and the leaves all fall, the woody plants are less directly affected by coal burning during the heating period. However, since the pine leaves do not fall and can also be respired, we may speculate that the pine leaves are affected by coal during the heating period, resulting in increased mercury content.

### 4.4. Risk Assessments

4.4.1. Evaluation of Soil Mercury Pollution Levels in Changchun City through the Geo-Accumulation Index ($I_{geo}$)

In this study, the descending-order table and the distribution chart of the soil accumulation index were obtained by means of the soil accumulation pollution index.

Based on the experimental results, it was found that, when compared with Zhengzhou City, the geological accumulation index of Changchun City was generally high, and the $I_{geo}$ value exceeded 5. This value indicated that some of the soil levels in Changchun City should be classified as severely polluted. Meanwhile, the maximum $I_{geo}$ value in Zhengzhou City was only 2.09, indicating moderate soil pollution.

As shown in Figure 7, the soil mercury pollution was serious in the western part of Changchun City. In addition, some contamination was also found in the northeastern section. This may be due to the presence of industrial parks, thermal power plants, and medical machinery factories in these areas, which potentially contributed to the relatively high levels of mercury detected in the soil. Previous studies have shown that industrial parks, thermal power plants, and medical machinery factories are major sources of mercury pollution. In addition, some contamination was detected in the northeastern section of the study area due to the dominant wind direction (southwest winds). The high levels of mercury content may also have been related to the existence of more logistics parks in the northeastern part of the city, and their cumulative index was much higher than that of the other locations in the study area. As a result, it was indicated that logistics transportation also significantly contributes to mercury pollution in urban soil. The mercury content of all of the in Changchun is above 0.052 mg/kg. According to the soil application function and protection objective, the soil standard classification can be divided into three categories. The primary standard is to protect the natural ecology of the region, and the mercury content standard is less than or equal to 0.15 mg/kg. It can be seen that more than half of the soil mercury content index in Changchun meets the primary soil standard, and the remaining soil mercury content meets the secondary and tertiary standards. The tertiary mercury pollution level of soil is not high, but it still needs to be controlled; otherwise, it will threaten the safety of the ecological environment.

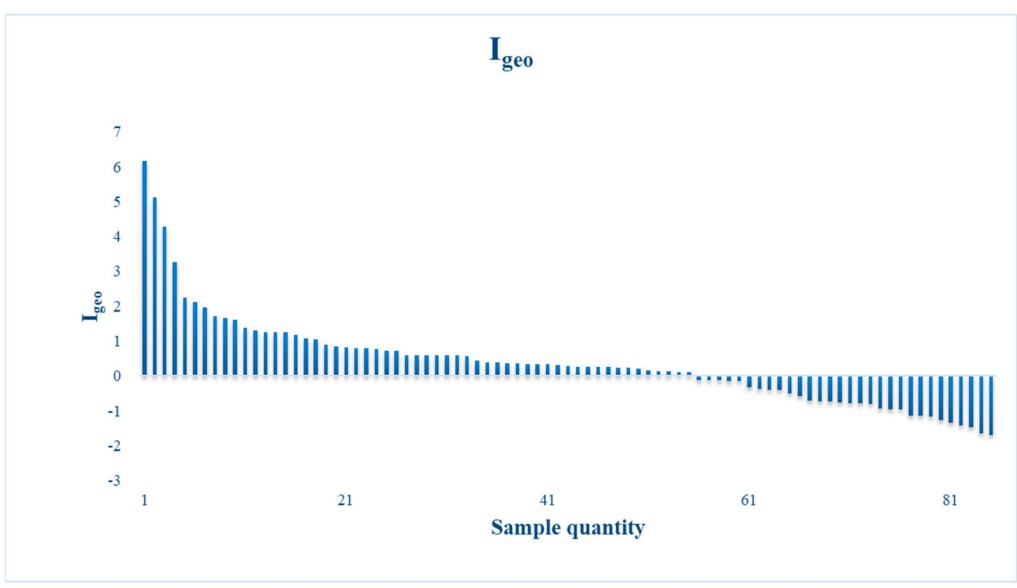

**Figure 7.** Descending chart of the $I_{geo}$.

4.4.2. Evaluation of Soil Mercury Pollution Levels in Changchun City Using the Potential Ecological Risk Index (Er)

Based on the characteristics of heavy metals and their environmental behaviors, the potential ecological risk index method is proposed to evaluate soil heavy metal pollution in soils or sediments from the perspective of sedimentology. An Er value of 40 is the limit value of potential ecological risk assessments. In the northwest (Er = 4296.4) and southeast (Er = 238.4) of Changchun, more than 66.6% of the sampling points exceeded this value, indicating an extremely serious ecological risk. The reason for these high risk values is that the land-use types in these two places are mainly industrial parks; industrial wastes and other pollutants lead to an increase in the soil mercury pollution content. According to the evaluation criteria of potential ecological risks, 15% of the sampling sites in Changchun had high potential ecological risks, 23.8% had moderate potential ecological risks, and 33.4% had low ecological risks. Therefore, this study found that the ecological risk level of soil mercury pollution was high in Changchun.

**5. Conclusions**

Based on our examination of common plant deciduous leaves and mercury pollution levels in the topsoil at 100 different sampling points in Changchun City, the following conclusions can be drawn.

It was determined that different plants have different mercury absorption and accumulation abilities, among which *Ussuri white* Maxim (0.0755 mg/kg) was observed to have a higher overall mercury accumulation ability. The mercury content levels in the leaves of the same tree species varied greatly in the different examined functional areas, with the industrial areas having the highest mercury content in the leaves. Meanwhile, the plants in the residential and park areas had lower levels of mercury pollution. In addition, transportation was determined to be an important factor affecting the mercury content levels in plant leaves. The distribution characteristics of soil mercury pollution in Changchun City were also found to be related to the different functional areas, and the surface soil mercury content in the industrial areas was relatively high. However, no significant correlation was observed between the mercury content in the plant leaves and the mercury content in the soil. This study's results indicated that the mercury pollution from plant waste in the various districts of Changchun City had not reached the level of severe pollution, with only the industrial areas experiencing moderate pollution. The other areas were determined to fall within the mild pollution range.

**Author Contributions:** Conceptualization, G.Z.; methodology, G.Z.; software, Z.Z.; validation, H.Z.; formal analysis, J.P.; investigation, J.P.; resources, G.Z.; data curation, J.Z.; writing—original draft preparation, J.P. and M.C.; writing—review and editing, Z.Z.; visualization, G.Z.; supervision, Z.W.; project administration, Z.W.; funding acquisition, Z.W. All authors have read and agreed to the published version of the manuscript.

**Funding:** This study was funded by the Chinese National Natural Science Foundation of China (Grant No. 31230012, 31770520); the Fundamental Research Funds for the Central Universities (No. 134-135132028); the Science and Technology Research Project of Jilin Provincial Education Department (No. JJKH20231316KJ); and the Chinese Postdoctoral Science Foundation (2021M700496).

**Institutional Review Board Statement:** Not applicable.

**Informed Consent Statement:** Not applicable.

**Data Availability Statement:** The data presented in this study are available on request from the corresponding author.

**Acknowledgments:** We are grateful to the Key Laboratory of Vegetation ecology of the Ministry of Education for its help and support.

**Conflicts of Interest:** The authors declare no conflict of interest.

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
