# Peer review of "Risk Assessments of Plant Leaf and Soil Mercury Pollution in Different Functional Areas of Changchun City"

_forests, doi:10.3390/f14061108_

Round 1

Reviewer 1 Report

This material is of scientific and practical interest, brings new data into scientific use. However, there are several comments that are reflected in the attached file.

Reviewer 2 Report

Heavy metals are added to the soil from various sources and their presence in the soil no doubt possess potential ecological risk and health hazard to human and animals via direct exposure or food chain. The present manuscript entitled “Pollution Characteristics and Risk Assessments of Plant Leaves and Soil Mercury Pollution in Different Functional Areas of Changchun City” is a quality research conducted by the authors. Hg is one of the top noxious pollutants that damage the environment as well as the living system. I appreciated the hard work and research concerns of the authors. However, despite the importance of the idea, the manuscript language in most section is confused and seems very poor, although I am not native to English. The results and discussion portions are intermixed; the authors need to properly arrange the results and discussions. The bulk of the tables and figures in results and then in discussion are very confusing. Moreover, I highlighted some changes in different sections of the manuscript, which may further aid in the quality of the manuscript. Therefore, I will recommend resubmission after major revision with thorough text filtration for improvement.

Title: Pollution characteristics… not understandable…. Please revise the title.. I would suggest delete the term pollution characteristics

Abstract

The abstract is very hard to understand, and difficult to come to some conclusions. It is because the abstract lacks the need for research, proper aims, and method—the abstract needs thorough revisions for a clear understanding of the readers. Some specific comments

- Line 16-17. Revises please hard to understand

-Line 18. What do you mean by deciduous. Did you mean deciduous plant species???

-Line 19. Elaborate please

-Line 22. Italicize please

-Line 22. Where are the other plants?

-Line 27. How it can be concluded

Line 28. How it can be said,, did you have any results or value to evaluate the risk

-Proper Conclusion missing

Introduction

The introduction is to the point but need language improvement in the sentence structure and also looks out for topographic mistakes. Many repetitive and extra preposition in the sentences. Some specific comments are

-Line 32. Delete pls.

-Line 38. Did u mean earth mental?

-Line 44-52. Poor language

Line 76. Delete Please

Line 97. goal.. or objectives…Goal is a very broad term

Line 104. Delete please.

Material and method

Well and comprehensive written but need topographic correction. In addition, it lacks informations on the soil, electrical conductivity, pH and and nutrient like organic matter which play critical role in the availability and accumulation of the heavy metals.

-Line 120. Semi climate meaning…….

Line 123. Very low....please confirm

Line 126. Delete Please

Line 135-36. is it needed after points area are mentioned

Line 137. Figure 1, and

Line 137. is it binomial, if yes italicize please

Line 154. Correct pls

Results

The results are interested, however, it can be better elaborated if more statistical tools are used i.e. ANOVA and post-hoc tests

-Line 208. Italicize the binomial please

- Line 254-256. Where is the correlation table?

- Line 262-266. More like discussion

- A table may better evaluated the ecological risk index and geo-accumulation index

Discussion

Mostly results are included please look the annotated version attached

No discussion found it’s the same results, Please add proper discussion justifying and contrasting and comparing your results with other works

Conclusion

The conclusion is the repetition of the results. The conclusion needs to be refined and provide recommendations for further studies. In addition try to reduce the bulk of result repetition in conclusion.

Round 2

Reviewer 2 Report

Line 2. Remove “The” from the title

Line 23. An RA… add n after A

Line 39 and 39. Delete “In conclusion, the distribution of mercury in plant 38

litter and soil in Changchun City is strongly affected by human activities.

 Line 154. Already corrected during language editing
